# Effects of Temperature and Pressure on Corrosion Behavior of HVOF-Sprayed Fe-Based Amorphous Coating on the Mg-RE Alloy for Dissolvable Plugging Tools

**DOI:** 10.3390/ma16031313

**Published:** 2023-02-03

**Authors:** Yijiao Sun, Hongxiang Li, Jun Yang, Jishan Zhang

**Affiliations:** 1State Key Laboratory for Advanced Metals and Materials, University of Science and Technology Beijing, Beijing 100083, China; 2National Engineering Laboratory for Exploration and Development of Low-Permeability Oil & Gas Fields, Xi’an 710021, China; 3Changqing Downhole Technology Company, Chuanqing Drilling Engineering Co., Ltd., Xi’an 710021, China

**Keywords:** dissolvable magnesium alloys, Fe-based amorphous coatings, high velocity oxygen-fuel (HVOF) spraying, corrosion resistance, high temperature and high pressure

## Abstract

To retard the degradation of the magnesium alloys for dissolvable ball seats, Fe-based amorphous coatings were deposited on dissolvable Mg-RE alloy substrates using high velocity oxygen-fuel spraying technology. The results show that the Fe-based amorphous coatings possess low porosity (0.82%) and high amorphous contents (91.4%) and their corrosion resistance decreases with the increase of temperature or pressure. However, with the help of Fe-based amorphous coatings, the degradation time of dissolvable Mg-RE alloy has been significantly prolonged. In particular, the service life of coated Mg-RE alloy exceeds 360 h at temperatures below 50 °C and reaches 87 h at 120 °C and 80 atm. Under high temperature and high pressure, the compactness of passive films decreases and the chemical activities of ions and metal elements increase, leading to the degradation of corrosion resistance of Fe-based amorphous coatings. In long-term corrosion, the crystallized splats are prone to corrosion because of the multiphase structures. The corroded crystallized splats are connected to the inevitable pores by the corroded intersplat regions, resulting in the formation of corrosion channels and the corrosion failure of coatings. This study provides a useful guidance for the corrosion protection of dissolvable plugging tools made of magnesium alloys.

## 1. Introduction

With the increase in demand for energy sources and the relative scarcity of conventional hydrocarbon resources, unconventional hydrocarbon resources have captured extensive attention for their abundant reserves [1,2,3]. Ball-activated multistage fracturing technology, one of the most efficient methods for the exploitation of unconventional hydrocarbon reservoirs, has been applied in many oil and gas fields. In this technology, the ball seats with various internal diameters are installed in horizontal wellbores and the staged fracturing operations are performed in turn with the cooperation of fracturing balls [4,5]. To further reduce the production costs and construction duration, new ball seats made of dissolvable magnesium alloys have been designed in recent years. Compared with the conventional ball seats, dissolvable ball seats can be degraded without drilling after fracturing operations. However, the environment of horizontal wells is extremely harsh and the dissolvable ball seats have to withstand the long-term corrosion of groundwater and fracturing fluids under high temperatures (50~150 °C) and high pressures (>50 MPa) [6,7,8]. Due to the high corrosion rates, the dissolvable ball seats are always destroyed before fracturing operations, leading to the leakage of high-pressure fluids and the abnormal operation of sliding sleeves. Therefore, a high-performance protective coating should be developed on the dissolvable ball seats to restrain their degradation and ensure their reliable service before the formal fracturing operation.

The normal techniques for fabricating the protective coatings on magnesium alloys include chemical conversion [9,10], micro-arc oxidation [11,12], anodizing [13], organic coating [14,15], electroless plating [16], electroplating [17], thermal spraying [18] and cold spraying [19]. Nevertheless, because of the lack of thickness, high porosity or low adhesion strength, the majority of them are unable to withstand corrosion under high temperature and high pressure. High velocity oxygen-fuel (HVOF) spraying, as a thermal spraying technology, has been widely used in the field of metal protection. In the process of HVOF spraying, the low temperature and supersonic velocity of in-flight molten droplets can markedly suppress the oxidation of droplets, the generation of pores and the performance deterioration of substrates [20]. Besides, the capacity to fabricate thicker coatings can effectively avoid the existence of through-porosity and further improve the long-term corrosion resistance of coatings [21]. Thus, the HVOF-sprayed coatings are expected to decrease the degradation rate of dissolvable magnesium alloy substrates and improve the service life of dissolvable ball seats, especially in the extreme downhole environment.

There are various HVOF-sprayed coatings on magnesium alloys, such as Al coatings [22], 316L coatings [23,24], SiC-Al coatings [25,26], WC-Co coatings [27], hydroxyapatite coatings [28,29] and Fe-based amorphous coatings (AMCs) [30,31,32,33]. Among these coatings, the AMCs attract more and more attention due to their superior corrosion resistance. Currently, HVOF-sprayed AMCs have been prepared for the corrosion protection of commercial magnesium alloys, such as AZ31, AZ61, WE43 and LA141 magnesium alloys [30,31,32,33]. Under normal temperature and pressure, the corrosion current densities of the coated magnesium alloys in 3.5 wt.% NaCl solution are one to two orders of magnitude lower than those of the bare magnesium alloys. In addition, the bottleneck problem on the low adhesion strength caused by performance difference has been solved via introducing a Ni60/NiCrAl interlayer, which further expands the application prospects of AMCs on magnesium alloys [30,33].

According to the recent studies, the superior corrosion resistance of HVOF-sprayed AMCs is attributed to their amorphous structure, abundant passivity promoters, plentiful dissolution moderators and high compactness [34,35]. The single amorphous phase can avoid the occurrence of galvanic corrosion and the formation of uneven passive films, so the amorphous content is an important indicator to evaluate the performance of AMCs. The Cr element, as a passivity promoter, is the main element in improving the corrosion resistance of AMCs since its hydroxides and oxides are the most important components of dense passive films. The dissolution moderators, such as Mo, Nb and W elements, will not only become oxides to improve the compactness of passive films, but also promote the enrichment of Cr element in passive films [35]. Unfortunately, the HVOF-sprayed AMCs are not perfect and their defects contain crystalline phases, Cr-depleted intersplat regions and inevitable pores [34]. Crystallization not only results in the non-uniform distribution of passive films, but also leads to potential differences among various phases. Thus, generally speaking, crystallization is detrimental to the corrosion resistance of AMCs. Due to the oxidation of metal droplets in the process of HVOF spraying, the intersplat regions lack Cr element and the intersplat regions, as an anode, are susceptible to micro-galvanic corrosion [36]. In long-term corrosion, the inevitable pores will be connected with each other by the corroded intersplat regions, leading to the formation of corrosion channels [37]. At present, the corrosion behaviors of AMCs under ordinary working conditions have been investigated, but there is little research on the corrosion resistance and corrosion mechanism of AMCs under high temperature and high pressure. Moreover, though HVOF-sprayed AMCs have been used for the corrosion protection of commercial magnesium alloys, it is still unknown whether AMCs can effectively protect the dissolvable magnesium alloys and extend the degradation time of dissolvable ball seats in the extreme downhole environment. Therefore, it is necessary to investigate the corrosion behaviors of the AMCs on dissolvable magnesium alloy substrates at various temperatures and pressures.

With this aim, AMCs were firstly prepared on the dissolvable Mg-Gd-Y-Zn-Cu alloy (Mg-RE alloy) via HVOF spraying technology. At various temperatures and pressures, the corrosion resistance of the AMCs and the degradation time of coated Mg-RE alloy were studied systematically. To study the corrosion mechanism of AMCs under high temperature and high pressure, the passive films and corrosion morphologies of AMCs after immersion were also analyzed in detail.

## 2. Materials and Methods

Atomized Fe-based amorphous powders (Fe_45_Cr_23_Mo_10_Si_9_C_9_B_4_, at.%) and Ni60 powders (Ni_50_Cr_17_Mo_1.6_Fe_3.7_Cu_1.7_Si_6_B_17.3_C_2.7_, at.%) were adopted to prepare the AMCs with Ni60 interlayers using a HVOF spraying system (Lijia thermal spraying machinery Co., Zhengzhou, China). Mg-Gd-Y-Zn-Cu alloy (Mg-RE alloy) with both high strength and high degradation rate was selected as the substrate. Mg-RE alloy plates with a dimension of 200 mm × 200 mm × 10 mm were cleaned, degreased and sandblasted before spraying.

Phase analyses of Fe-based amorphous powders and AMCs were carried out using an X-ray diffractometer (XRD, Rigaku Corporation, Tokyo, Japan) with Cu Kα radiation. The detected angle (2θ) was scanned from 20° to 80° at a scanning speed of 4°/min. Thermostability and exothermic enthalpy of the powders and coatings were measured using a differential scanning calorimeter instrument (DSC, TA Instruments, New Castle, DE, USA) at a heating rate of 10 K/min. Equipped with energy dispersive spectroscopy (EDS), scanning electron microscopy (SEM, Carl Zeiss AG, Oberkochen, Germany) was used to obtain the morphology and chemical composition of the powders and AMCs. The accelerating voltage was selected as 15 kV and the working distance was about 8 mm. Transmission electron microscopy (TEM, Titan, FEI Company, Hillsboro, OR, USA) was used to observe the microstructure and analyze the chemical composition of the AMCs.

Electrochemical performances of the AMCs at various temperatures and pressures were evaluated in an autoclave equipped with a three-electrode system. Inert nitrogen was selected as pressurized gas and 3 wt.% KCl aqueous solution was used as the simulated groundwater and fracturing fluid. All the samples for electrochemical corrosion tests were embedded with the heat-resistant epoxy resin and only a mirror-finished area of 1 cm^2^ was exposed in corrosive solution. After the open circuit potentials were stable, the potentio-dynamic polarization measurements, electrochemical impedance spectroscopy (EIS) and Mott-Schottky measurements were performed using an electrochemical workstation (Corrtest Instruments Corporation, Wuhan, China). The scanning rate of potentio-dynamic polarization tests was set to 0.5 mV/s and the Tafel curves were analyzed by Versa Studio software. In the EIS tests, the amplitude of the sinusoidal voltage signal was set to 20 mV and the data was fitted using Zview software. Mott-Schottky measurements were performed in the potential range of −0.6~1.0 V_Ag/AgCl_ and the scanning rate and frequency were 25 mV/s and 1 kHz, respectively.

Immersion tests of the bare Mg-RE alloy and the coated Mg-RE alloy at various temperatures and pressures were carried out in the autoclave. The test samples with the size of 20 mm × 20 mm × 10 mm were coated by heat-resistant epoxy resin and only an area of 4 cm^2^ was exposed. The mass variation of samples with the immersion time were recorded. After immersion for 10 days, the composition and valence states of the passive films formed on AMCs were detected by X-ray photoelectron spectroscopy (XPS, Kratos Analytical, Manchester, UK) with Al Kα excitation. The Avantage software was used to fit and analyze the XPS data. Both the electrochemical tests and immersion tests were repeated at least three times to ensure the data integrity.

## 3. Results and Discussion

### 3.1. Characterization of the AMCs on Dissolvable Mg-RE Alloy

The scanning electron microscope-secondary electron (SEM-SE) image of Fe-based amorphous powders and the cross-sectional scanning electron microscope-backscattered electron (SEM-BSE) images of AMCs are shown in Figure 1a–c. The Fe-based amorphous powders used for HVOF spraying possess smooth surface and spherical shape and their diameters range from 5 μm to 50 μm. AMCs with a thickness of 370 μm are prepared on the dissolvable Mg-RE alloy substrates. To improve the adhesion strength, Ni60 interlayers have been deposited on the substrates in advance [30]. The AMCs whose porosity is about 0.82% are composed of stacked splats, inevitable pores and a few unmelted powder particles. Besides, there are distinct boundaries, termed intersplat regions, among these splats. Figure 1d,e present the XRD patterns and DSC curves of the Fe-based amorphous powders and AMCs, respectively. According to the XRD patterns, only a broad diffraction hump appears in the 2θ range of 34° to 56°, suggesting that both the Fe-based amorphous powders and the AMCs have high amorphous content. From the DSC curves, it can be seen that there are four obvious exothermic peaks caused by crystallization in the range of 600 °C to 875 °C and the crystallization enthalpies of Fe-based amorphous powders and AMCs are 174 J/g and 159 J/g, respectively. By dividing the crystallization enthalpy of powders by that of AMCs, the amorphous content of AMCs can be calculated as 91.4%.

The TEM images, selected area electron diffraction (SAED) patterns and EDS line scanning analyses of the AMCs are illustrated in Figure 2. Consistent with the previous conclusion, the AMCs are formed by the accumulation of splats and the morphology of intersplat regions is significantly different from that of splats. The only diffraction halo on the SAED pattern of Zone A indicates that these splats exhibit an amorphous structure. Based on the EDS line scanning profiles, the distribution of Fe, Cr, Mo and Si elements in these splats is uniform, but the intersplat regions are rich in O element. The high-resolution electron microscope (HREM) image shows that, as in the amorphous splats, the arrangement of atoms in the intersplat region is disorderly. Thus, it can be inferred that these oxides distributed among splats are also amorphous and they are mainly generated from the surface oxidation of metal droplets during HVOF spraying [36]. As shown in Figure 2d, there are large numbers of fine crystalline phases in the splats and the crystallization can also be confirmed by the discontinuous diffraction rings on the SAED pattern of Zone C. Therefore, except for the amorphous splats and O-rich intersplat regions, the AMCs also contain a small number of crystallized splats.

### 3.2. Corrosion Resistance of the AMCs on Dissolvable Mg-RE Alloy

Figure 3a,b display the polarization curves of the AMCs in 3 wt.% KCl solution under various temperatures and pressures and the corrosion data determined from the polarization curves are listed in Table 1. The corrosion potentials (E_corr_) of the AMCs decrease with the increase of temperature or pressure, which is consistent with other research [38,39]. Regardless of pressure, the corrosion current densities (I_corr_) of the AMCs increase with increasing temperature. At the same temperature, the corrosion current densities of the AMCs at normal pressure are smaller than those at high pressure. Therefore, no matter whether from the point of view of thermodynamics or kinetics, the corrosion resistance of the AMCs will be deteriorated with the increase of temperature or pressure. As shown in Figure 3c, the corrosion current densities of the AMCs as a function of temperature can be fitted to the Arrhenius equation and the equation is as follows [39]:(1)$$lnk=\frac{-E_{a}}{RT}+C$$
where *k* represents the corrosion current densities of the AMCs, *C* stands for a constant associated with chemical reactions, *E_a_* donates the activation energy of the corrosion reactions, *R* refers to the gas constant and *T* is the Kelvin temperature. Through fitting and analysis, there is an obvious linear relationship between the natural logarithm of corrosion current densities and the multiplicative inverse of temperature at both normal pressure and high pressure, suggesting that the corrosion of the AMCs is actually a thermally activated process when the temperature ranges from 20 °C to 120 °C. It is reported that the activation energies of the diffusion-controlled reactions in liquid phases are usually less than 41.84 KJ/mol and the activation energy of the corrosion reaction of 304 stainless steel in aqueous solution is about 14.66 KJ/mol [39,40]. In this work, the activation energies of the corrosion reactions at normal pressure and high pressure are calculated as 12.03 KJ/mol and 16.36 KJ/mol, respectively, which are within a reasonable range. In addition, the activation energy of the corrosion reaction at high pressure is higher than that at normal pressure, so the corrosion reaction of the AMCs at high pressure is more sensitive to temperature. In terms of passivation ability, there is a clear passivation region on each polarization curve, confirming that the AMCs still have good passivation ability regardless of temperature and pressure. The corrosion potentials and pitting potentials (E_pit_) under different temperatures and pressures are depicted in Figure 3d. Different from the variation of corrosion potentials, the pitting potentials of the AMCs decrease with increasing temperature, but the hydrostatic pressure of 80 atm has little influence on them. The variation of passivation current densities (I_pass_) with temperature and pressure is similar to that of corrosion current densities, i.e., the passivation current densities increase with the increase of temperature or pressure. In previous studies, it is reported that temperature and pressure lead to the deterioration of corrosion resistance and pitting resistance of most alloys, such as 316 stainless steel, 304 stainless steel, Alloy 690, Fe-20Cr alloys and 2205 duplex stainless steel [39,40,41,42,43]. In the present work, although the corrosion resistance of AMCs decreases with increasing temperature or pressure, the hydrostatic pressure has little effect on pitting resistance of AMCs and the AMCs still exhibit superior pitting resistance under high temperature and high pressure. In particular, the pitting potential of the AMCs under a temperature of 120 °C and pressure of 80 atm is about 0.812 V_Ag/AgCl_.

The Nyquist plots, Bode impedance magnitude plots and Bode phase angle plots of the AMCs at normal pressure are shown in Figure 4a–c. It seems that the AMCs have only one capacitive loop from the Nyquist plots, but they have two capacitive loops in fact, according to the Bode phase angle plots. The capacitive loops of the AMCs at 20 °C show the largest diameters and the diameters of capacitive loops decrease obviously with increasing temperature. The Nyquist plots, Bode impedance magnitude plots and Bode phase angle plots of the AMCs at high pressure are shown in Figure 4d–f. At a hydrostatic pressure of 80 atm, the AMCs still possess two capacitive loops and, the higher the temperature is, the smaller the diameters of the capacitive loops. Furthermore, the diameters of the capacitive loops of the AMCs at high pressure are slightly smaller than that at normal pressure when the temperature is the same. Figure 4b,e display the equivalent circuit model of the AMCs accompanied with the schematic diagram. The model of AMCs is composed of solution resistance (R_s_), the sum of pore resistance and coating resistance (R_c_), non-ideal coating capacitance (CPE_c_), the charge transfer resistance of the solution/AMC interfaces (R_t_) and the non-ideal capacitance of the solution/AMC interfaces (CPE_dl_). The EIS fitted results for the AMCs under various temperatures and pressures are shown in Table 2. Generally speaking, R_c_ and R_t_ are always used to evaluate the corrosion resistance of coatings and, the larger the resistance is, the better the corrosion resistance of coatings [38,41]. It is found that the R_c_ and R_t_ decrease with the increase of temperature no matter whether at normal pressure or high pressure, but the pressure of 80 atm only reduces the R_c_ and R_t_ slightly at the same temperature. Therefore, consistent with the results of polarization curves, the corrosion resistance of AMCs will descend when the temperature or pressure increases and the influence of temperature on the corrosion resistance is larger than that of hydrostatic pressure. In aqueous solution, the increase of temperature will not only increase the diffusion rate of ions, but also accelerate the corrosion reactions at the solution/AMC interfaces, so the pore resistance and charge transfer resistance will decrease when the temperature increases. In terms of hydrostatic pressure, the chemical activities of ions and metal elements at the solution/AMC interfaces increase with the increase of pressure, leading to the promotion of corrosion reactions [44]. Thus, similar to temperature, the charge transfer resistance of the solution/AMC interfaces also decreases when the pressure increases to 80 atm.

### 3.3. Semiconductor Properties and Chemical Composition of Passive Films on the AMCs

In 3 wt.% KCl aqueous solution, passive films with semiconductor properties can be formed on the surface of AMCs and their performance heavily determines the corrosion resistance of AMCs. The Mott Schottky plots and carrier concentration of the passive films formed at various temperatures and pressures are illustrated in Figure 5. All the Mott-Schottky plots exhibit a linear region with positive slope and a linear region with negative slope; the former ranges from −0.2 to 0.5 V_Ag/AgCl_, while the latter ranges from 0.5 to 0.8 V_Ag/AgCl_. Therefore, regardless of temperature and pressure, the passive films formed on the AMCs are bipolar, that is, they have both n-type and p-type semiconductor properties. In general, the carrier concentration is an important basis for evaluating the compactness of passive films and the passive film with low carrier concentration always shows better compactness [38]. Based on the Mott-Schottky relations, the donor concentration (*N_D_*) and the acceptor concentration (*N_A_*) can be calculated via the following formulas [45]:(2)$$C_{sc}^{-2}=\frac{2}{eN_{D}\varepsilon \varepsilon_{0}}\left(E-E_{FB}-\frac{kT}{e}\right)\;\left(n\text{-}\mathrm{type}\right)$$
(3)$$C_{sc}^{-2}=\frac{-2}{eN_{A}\varepsilon \varepsilon_{0}}\left(E-E_{FB}-\frac{kT}{e}\right)\;\left(p\text{-}\mathrm{type}\right)$$
where C*_SC_* refers to the space charge capacitance, e denotes the electronic charge, *ε*_0_ stands for the vacuum permittivity, *ε* is the relative permittivity of passive film (15.6 in this work), *k* refers to the Boltzmann constant, *T* represents the Kelvin temperature, *E* denotes the film formation potential and *E_FB_* denotes the flat band potential. No matter whether under normal pressure or high pressure, both the N_D_ and N_A_ increase observably with the increase of temperature, indicating the decline of the compactness of passive films with increasing temperature. Moreover, the *N_D_* and *N_A_* decrease as the pressure increases to 80 atm, so the increase of pressure can also damage the compactness of passive films.

Figure 6 presents the Fe 3p, Cr 3p and Mo 4d spectra for the AMCs after 10-day immersion in 3 wt.% KCl solution under different temperatures and pressures. It is found that no matter what the temperature and pressure are, the types of double peaks fitted for the XPS spectra are same, suggesting that the temperature and pressure have no effect on the component type of passive films. However, the contents of different compounds in the passive films vary with temperature and pressure. The Fe-containing compounds in the passive films are mainly FeO, Fe_2_O_3_ and FeOOH. Under temperature of 20 °C and the pressure of 1 atm, the ratio of Fe_ox_/Fe_hy_ in the passive film is 2.16 and the Fe^2+^/Fe^3+^ ratio is 0.36. With the increase of temperature or pressure, the relative contents of iron oxides and Fe^2+^ increases significantly. When the temperature is 120 °C and the pressure is 80 atm, the ratio of Fe_ox_/Fe_hy_ in the passive film reaches 3.60 and the Fe^2+^/Fe^3+^ ratio increases to 0.61. According to the fitted results from Cr 3p spectra, the Cr-containing compounds in the passive films mainly contain Cr_2_O_3_ and Cr(OH)_3_. The ratio of Cr_ox_/Cr_hy_ in the passive film is 2.81 at normal temperature and pressure. The ratio of Cr_ox_/Cr_hy_ decreases to 1.54 when the pressure increases to 80 atm and the ratio of Cr_ox_/Cr_hy_ reaches 0.90 as the temperature increases to 120 °C. Under temperature of 120 °C and pressure of 80 atm, the ratio of Cr_ox_/Cr_hy_ is about 0.61. In terms of Mo element, the Mo-containing compounds in the passive films consist of MoO_2_ and MoO_3_. Different from the compounds of Fe or Cr, the temperature and pressure have little influence on the relative content of molybdenum oxides.

Chemical compositions of the passive films formed on AMCs under various temperatures and pressures are shown in Figure 7. With the increase of temperature or pressure, the content of Fe element in the passive films decreases, while the content of Cr element increases significantly. Besides, there is no obvious variation in the content of Mo element in the passive films regardless of temperature and pressure. It is reported that the Cr-containing compounds are more stable than the Fe-containing compounds and the Fe element in the passive film is more easily dissolved in a more hostile environment, leading to the reduction of the Fe/Cr ratio in the passive film [46]. For the oxides and hydroxides of Fe, the content of FeO hardly changes with the increase of pressure, but slightly decreases with the increase of temperature. However, the contents of Fe_2_O_3_ and FeOOH observably decrease with the increase of temperature or pressure. Therefore, the more aggressive environments, such as high temperature and high pressure, will promote the dissolution of Fe_2_O_3_ and FeOOH (Fe^3+^) at the passive film/solution interface, resulting in the increase of Fe_ox_/Fe_hy_ ratio and Fe^2+^/Fe^3+^ ratio. In terms of the oxides and hydroxides of Cr, with the increase of temperature or pressure, the content of Cr_2_O_3_ decreases but the content of Cr(OH)_3_ increases. For the oxides of Mo, the contents of MoO_2_ and MoO_3_ are not sensitive to the temperature and pressure. As is well known, the increase of hydroxides or the decrease of oxides will significantly reduce the compactness of the passive film [47,48]. In this work, the atomic percentage of hydroxides in the passive film formed at normal temperature and pressure is about 25.1%. The atomic percentage rises to 28.9% when the pressure increases to 80 atm and the atomic percentage rises to 38.7% as the temperature increases from 20 °C to 120 °C. At temperature of 120 °C and pressure of 80 atm, the atomic percentage of hydroxides reaches 44.2%. Therefore, consistent with the Mott-Schottky analysis, the compactness of passive films decreases with the increase of temperature or pressure and the influence of the temperature of 120 °C on passive films is obviously higher than that of the hydrostatic pressure of 80 atm.

### 3.4. Long-Term Corrosion and Degradation of the Coated Dissolvable Mg-RE Alloy

To evaluate the effective protective ability of the AMCs on dissolvable Mg-RE alloy substrates, the coated Mg-RE alloy was immersed in 3 wt.% KCl solution under various temperatures and pressures and the weight loss–time curves for the bare Mg-RE alloy and the coated Mg-RE alloy are shown in Figure 8. At normal pressure, the degradation time of the bare Mg-RE alloy at 20 °C, 50 °C, 90 °C and 120 °C is about 94 h, 46 h, 33 h and 24 h, respectively. Thus, the degradation rate of Mg-RE alloy increases with the increase of temperature. However, when the temperature is 20 °C or 50 °C, the mass of the coated Mg-RE alloy increases slightly with time and, the higher the temperature is, the more the mass increases. The mass increase of the coated Mg-RE alloy is not only due to the wetting and infiltration of corrosive solution, but also to the accumulation of corrosion products caused by the corrosion of the coating or substrate [49]. At the temperature of 90 °C and 120 °C, the mass of coated Mg-RE alloy increases at the initial stage of immersion and then abruptly decreases after a specific time. In addition, the mass increase of the coated Mg-RE alloy at 120 °C is faster than that at 90 °C. The sudden decline in the mass of coated Mg-RE alloy occurs at the immersion time of 203 h when the temperature is 90 °C and the immersion time is reduced to 118 h as the temperature is 120 °C. As mentioned before, the long-term corrosion makes the corrosive solution permeate the coating and reach the coating/substrate interface [37,49]. The continuous generation and accumulation of corrosion products results in the rupture of the AMCs and the exposure of the substrate. In other words, the immersion time when the mass decline takes place is actually the service life of the AMCs.

In terms of the corrosion at the pressure of 80 atm, the degradation time of the bare Mg-RE alloy at 20 °C, 50 °C, 90 °C and 120 °C is 101 h, 50 h, 37 h and 27 h, respectively. The corrosion rate of the bare Mg-RE alloy also increases with the increase of temperature at high pressure. However, it is worth noting that the corrosion rate of Mg-RE alloy at high pressure is lower than that at normal pressure. This phenomenon is related to the fact that high pressure can not only expedite the adsorption of [H] on the surface of Mg-RE alloy, but also impede the desorption of H_2_ from the surface of Mg-RE alloy [42]. At the temperature of 20 °C and 50 °C, the mass of the coated Mg-RE alloy also increases with the immersion time and same as the coated Mg-RE alloy at normal pressure, the coated Mg-RE alloy at high pressure is not degraded after immersion for 360 h. At the pressure of 80 atm, the service lives of the AMCs at 90 °C and 120 °C are 129 h and 87 h, respectively. The service life of the AMCs at high pressure is significantly lower than that at low pressure, indicating the degradation in the protective ability of AMCs at high pressure. To sum up, the AMCs can provide long-term corrosion protection for the Mg-RE alloy substrate, but the protective ability of AMCs will decline under high temperature and high pressure.

Figure 9 displays the corrosion morphology of the coated Mg-RE alloy after immersion for 5 days in 3 wt.% KCl solution under various temperatures and pressures. When the temperature is 20 °C or 50 °C, only a few inevitable pores and some boundaries among the splats can be observed on the surfaces of AMCs and the pressure of 80 atm has no effects on the surface morphology. However, as the temperature increases to 90 °C, some round corrosion products appear on the coating surfaces and these corrosion products are rich in O, Fe, Cr and Mo elements according to the EDS point analysis. Thus, it can be inferred that the pitting corrosion has occurred on the surfaces of AMCs. In addition, both the number and diameter of pittings at high pressure are larger than those at normal pressure, suggesting that the hydrostatic pressure of 80 atm reduces the corrosion resistance of AMCs. Under the temperature of 120 °C and the pressure of 2 atm, lots of splats on the surface of AMCs are corroded partially. When the pressure increases to 80 atm, not only the number of corroded splats increases, but also these splats are corroded completely. It is noteworthy that the corrosion only occurs on some specific splats and it does not extend to the surrounding uncorroded splats. Moreover, based on the corresponding EDS point analysis, the corrosion products are rich in Ni element except for Fe, Cr, O and Mo elements. Due to the absence of Ni element in the AMCs, the Ni-containing corrosion products comes from the corroded Ni60 interlayer, i.e., the corrosive solution has passed through the AMCs and penetrated into the Ni60 interlayers. In short, both high temperature and high pressure can promote the corrosion of the AMCs and the corrosion only takes place on some specific splats, which is quite different from that of most alloys [39,40,41,42,43].

### 3.5. Corrosion Mechanism of the AMCs on Dissolvable Mg-RE Alloy

The schematic diagrams for the corrosion mechanism of the AMCs on dissolvable Mg-RE alloy are shown in Figure 10. Due to the inherent shortcomings of HVOF spraying, not only amorphous splats but also a small number of crystallized splats exist in the AMCs. The passivity promoters and dissolution moderators, such as Cr and Mo elements, are distributed unevenly in the crystallized splats, resulting in the formation of uneven and flimsy passive films on their surfaces. In addition, there are obvious potential differences among the various phases and micro-galvanic corrosion is prone to take place in the crystallized splats [50]. Therefore, the pitting corrosion always occurs on the crystallized splats preferentially in the early stage of immersion corrosion and the corrosion gradually expands outward with the immersion time. After the crystallized splats are corroded completely, corrosion pits with the same size as the splats will be formed inside the AMCs and the corrosive solution begins to infiltrate into the AMCs. It has been reported that the intersplat regions are susceptible to corrosion because of the lack of Cr elements and they always connect to the pores to form corrosion channels in long-term corrosion [36,37]. However, there are few reports about the influence of the crystallized splats on the long-term corrosion behaviors of the Fe-based amorphous coatings. In the present study, the corroded intersplat regions serves as a bridge between pores and corrosion pits (corroded crystallized splats), leading to the formation of corrosion channels inside the AMCs. Through these channels, the corrosive solution passes through the whole AMCs and reaches the AMC/Ni60 interfaces. Because of the poor corrosion resistance, the Ni60 interlayers will be corroded and the Ni-containing corrosion products on the coating surfaces will be formed by the released Ni ions. As the corrosion progresses further, the Ni60 interlayer fails and the corrosive solution comes into contact with the Mg-RE alloy substrates. Because of the high chemistry/electrochemical activity, the dissolvable Mg-RE alloy substrates are rapidly corroded, resulting in large numbers of corrosion products at the Ni60/Mg-RE alloy interfaces. Ultimately, these corrosion products burst the coating and the dissolvable Mg-RE alloy substrates are fully exposed to the corrosive solution and rapidly degraded.

## 4. Conclusions

To ensure the normal use of the ball seats made of dissolvable magnesium alloys prior to fracturing operations, the AMCs are fabricated on dissolvable Mg-RE alloy substrates by HOVF spraying technology. The microstructure, corrosion resistance and corrosion mechanism of the AMCs are investigated systematically. The following conclusions can be drawn from this study.

(1)The AMCs on dissolvable Mg-RE alloy substrates possess low porosity (0.82%) and high amorphous contents (91.4%). In addition to the amorphous splats, inevitable pores and O-rich intersplat regions, the AMCs also contain a small number of crystallized splats.(2)The corrosion resistance of AMCs decreases with the increase of temperature or pressure, but the AMCs still have excellent pitting resistance. High temperature and high pressure not only reduce the compactness of the passive films formed on the coating surfaces, but also increase the chemical activities of ions and metal elements at the interfaces of solution/AMC.(3)Regardless of temperature and pressure, the degradation time of the coated Mg-RE alloy is significantly longer than that of the bare Mg-RE alloy. When the temperature is lower than 50 °C, the coated Mg-RE alloy will not be degraded within 360 h. At 120 °C and 80 atm, the degradation time of the coated Mg-RE alloy is more than 87 h, longer than that of the bare Mg-RE alloy (27 h).(4)In long-term corrosion, the crystallized splats in AMCs are corroded preferentially because of the uneven passive films and potential differences. The corroded intersplat regions act as a bridge to connect the pores and the corroded crystallized splats, resulting in the formation of corrosion channels and the degradation of dissolvable Mg-RE alloy substrates.(5)This work provides an effective solution for the corrosion protection of the plugging tools made of dissolvable magnesium alloys. However, the hydrostatic pressure of 80 atm is insufficient for real downhole environments. Research on the corrosion behaviors of AMCs and magnesium alloys at higher hydrostatic pressure or under actual operating conditions still need to be performed in the future.

## Figures and Tables

**Figure 1 materials-16-01313-f001:**
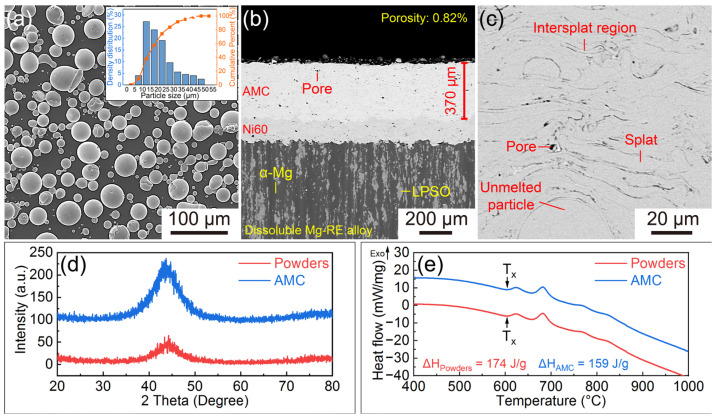
(**a**) SEM-SE image and size distribution of Fe-based amorphous powders; (**b**,**c**) Cross-sectional SEM-BSE images of the AMCs; XRD patterns (**d**) and DSC curves (**e**) of the Fe-based amorphous powders and AMCs.

**Figure 2 materials-16-01313-f002:**
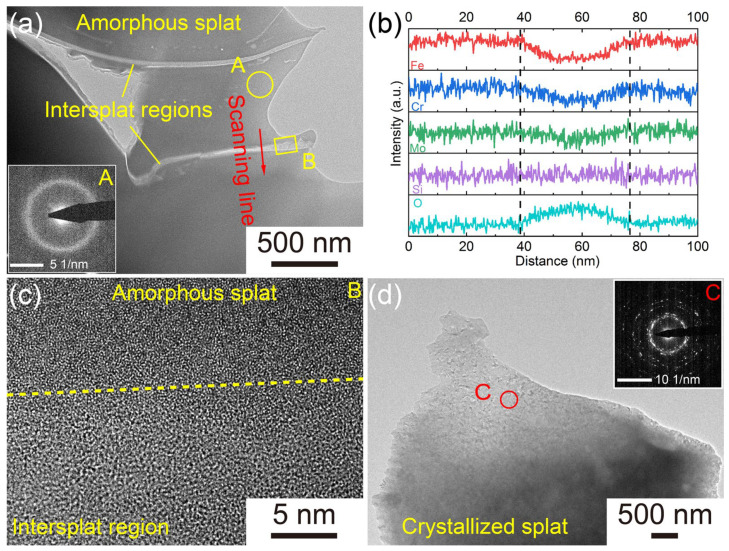
(**a**) TEM image of the amorphous splats in AMCs; (**b**) EDS line scanning profiles of the intersplat region marked in (**a**); (**c**) HREM image of Zone B (the intersplat region) marked in (**a**); (**d**) TEM image of the crystallized splats in AMCs; The insets in (**a**,**d**) are the SAED patterns of Zone A and C, respectively.

**Figure 3 materials-16-01313-f003:**
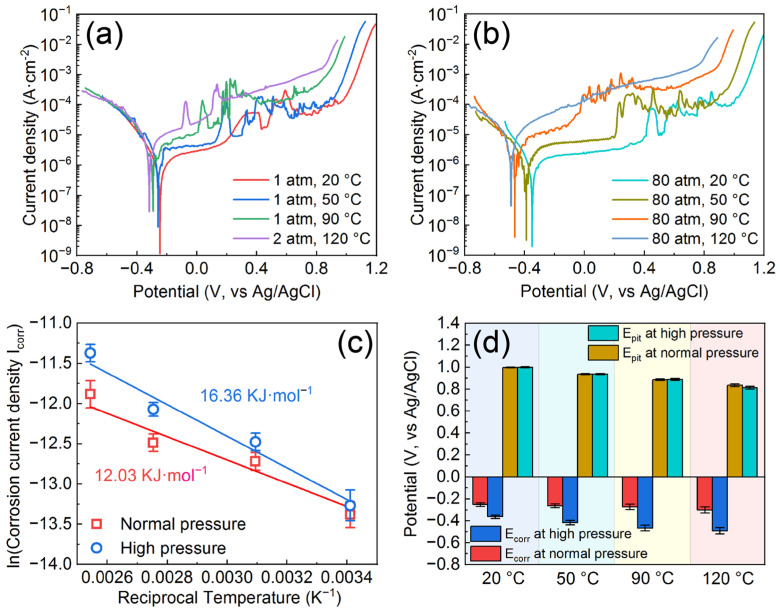
Potentio-dynamic polarization curves of the AMCs under normal pressure (**a**) and high pressure (**b**); (**c**) Arrhenius plots of the corrosion current densities of the AMCs at different pressures; (**d**) Corrosion potentials and pitting potentials of the AMCs under various temperatures and pressures.

**Figure 4 materials-16-01313-f004:**
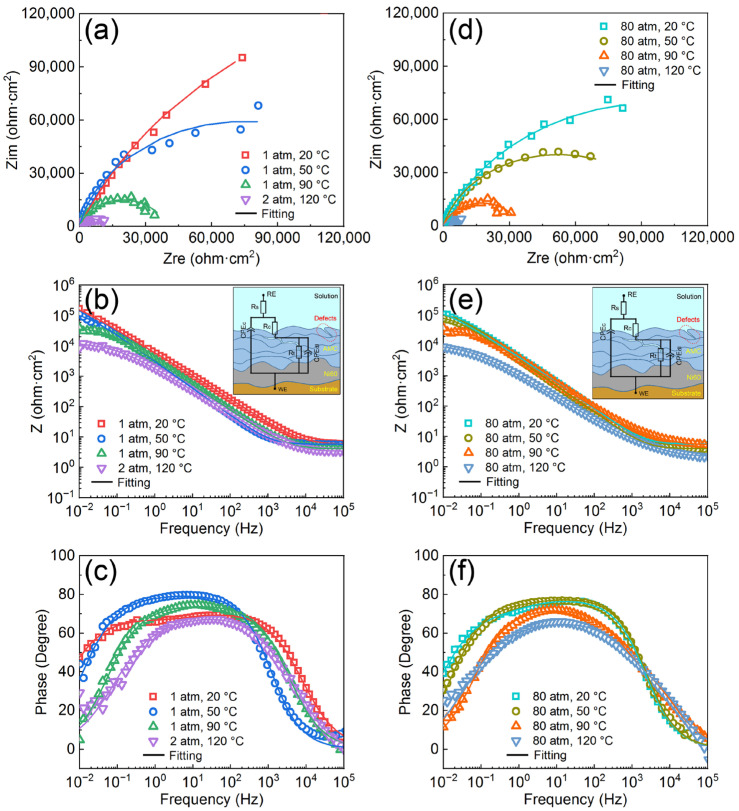
Nyquist plots (**a**), Bode impedance magnitude plots (**b**) and Bode phase angle plots (**c**) of the AMCs at normal pressure; Nyquist plots (**d**), Bode impedance magnitude plots (**e**) and Bode phase angle plots (**f**) of the AMCs at high pressure. The insets in (**b**,**e**) are the equivalent circuit model and schematic diagram of the AMCs.

**Figure 5 materials-16-01313-f005:**
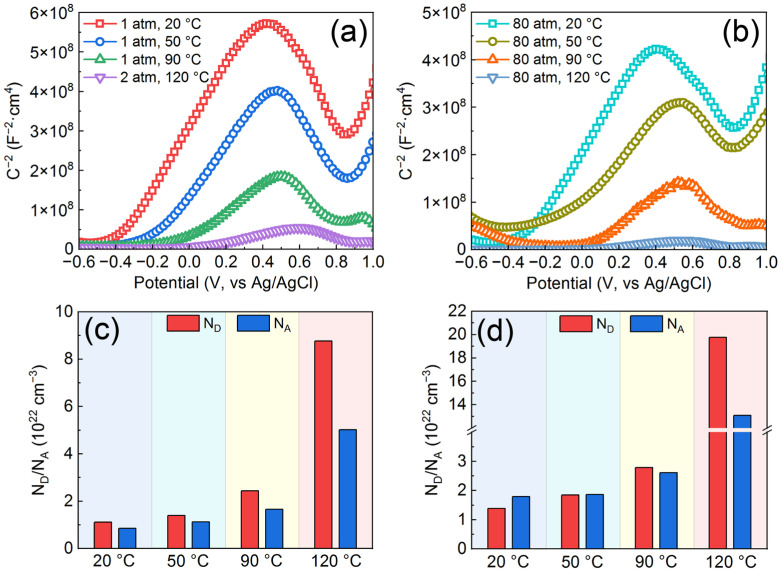
Mott-Schottky plots for the passive films formed on the AMCs: (**a**) at normal pressure, (**b**) at high pressure; Carrier concentration of the passive films formed on the AMCs: (**c**) at normal pressure, (**d**) at high pressure.

**Figure 6 materials-16-01313-f006:**
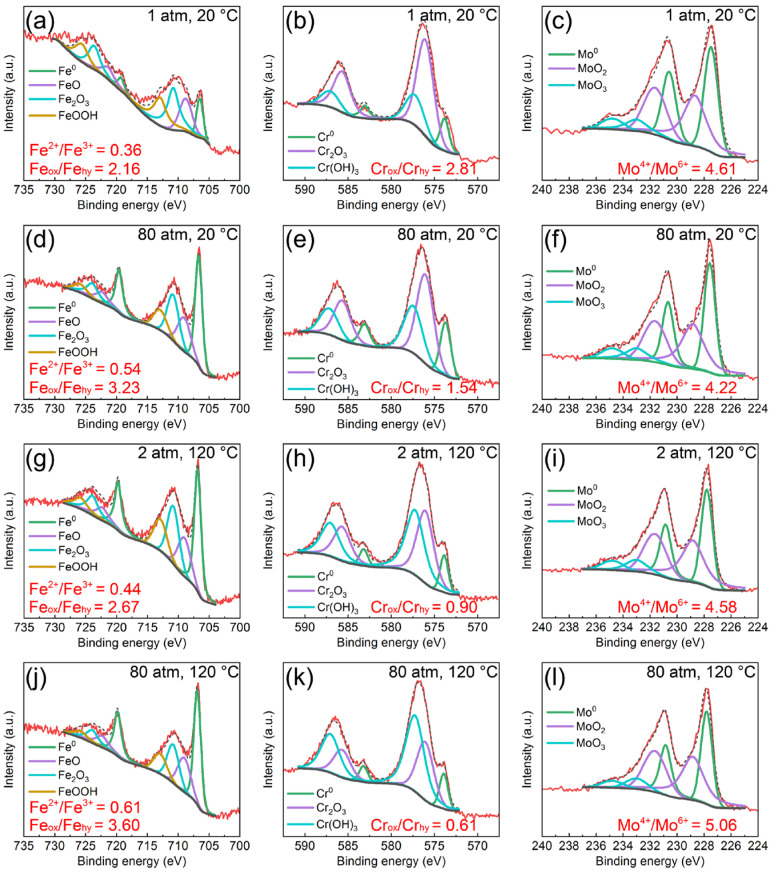
Fe 3p, Cr 3p and Mo 4d spectra for the AMCs after 10-day immersion in 3 wt.% KCl solution under different temperatures and pressures: (**a**–**c**) 1 atm and 20 °C, (**d**–**f**) 80 atm and 20 °C, (**g**–**i**) 2 atm and 120 °C, (**j**–**l**) 80 atm and 120 °C.

**Figure 7 materials-16-01313-f007:**
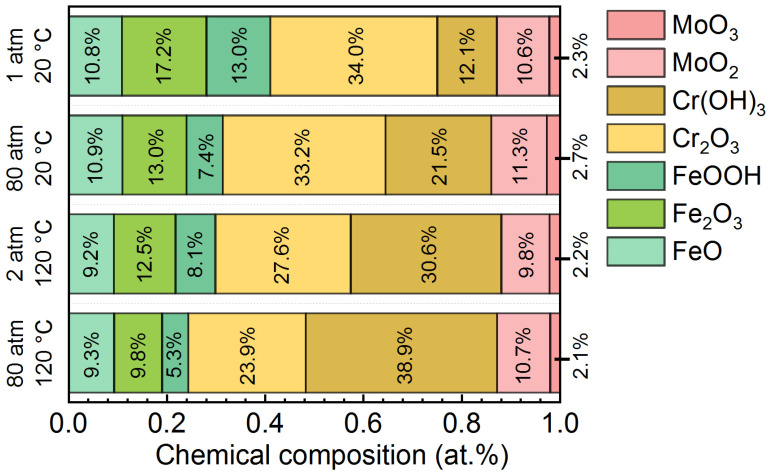
Chemical compositions of the passive films formed on the AMCs under different temperatures and pressures.

**Figure 8 materials-16-01313-f008:**
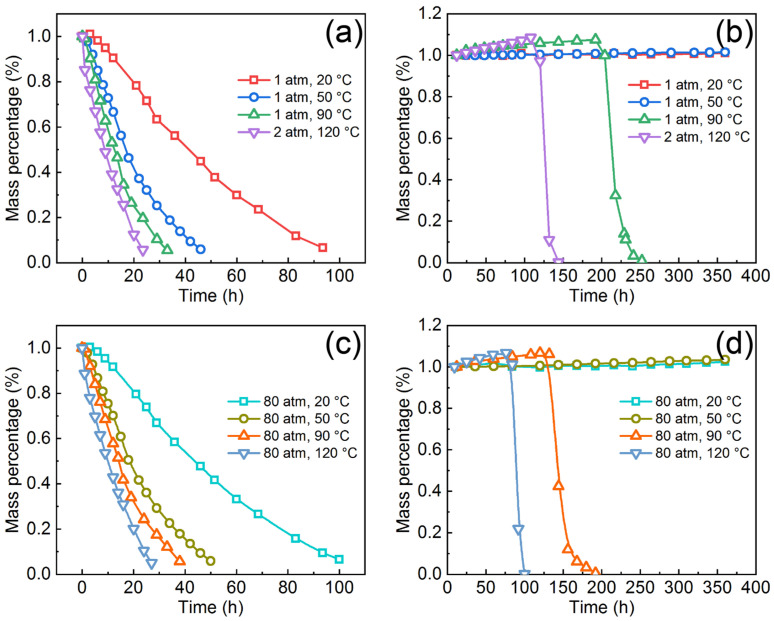
Weight loss-time curves for the bare Mg-RE alloy and the coated Mg-RE alloy in 3 wt.% KCl solution: (**a**,**b**) at normal pressure, (**c**,**d**) at high pressure.

**Figure 9 materials-16-01313-f009:**
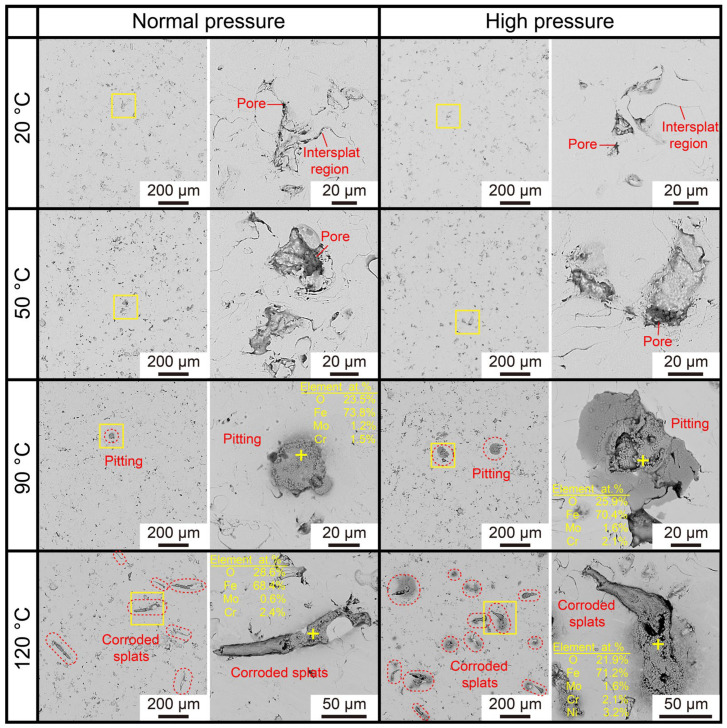
SEM-BSE micrographs of the coated Mg-RE alloy after 5-day immersion in 3 wt.% KCl solution under various temperatures and pressures.

**Figure 10 materials-16-01313-f010:**
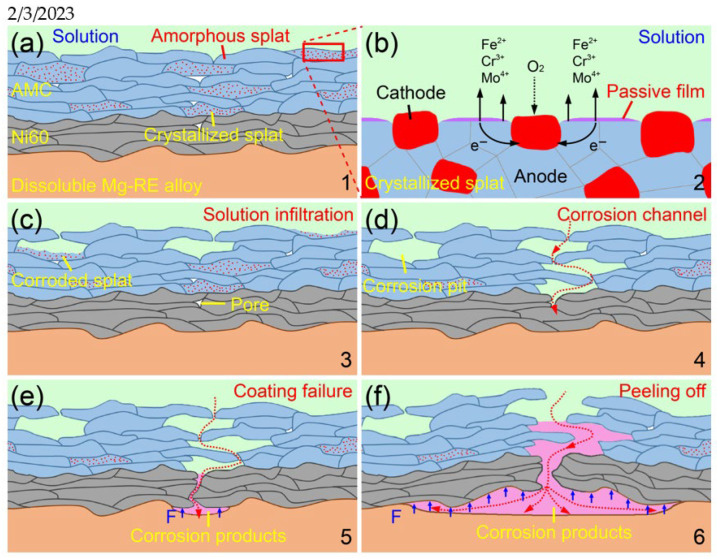
Schematic diagrams showing the corrosion mechanism of the AMCs on dissolvable Mg-RE alloy substrates: (**a**) uncorroded AMCs, (**b**) micro-galvanic corrosion of crystallized splats, (**c**) solution infiltration, (**d**) formation of corrosion channels, (**e**) coating failure and (**f**) peeling off of AMCs.

**Table 1 materials-16-01313-t001:** Corrosion data determined from the potentio-dynamic polarization curves.

Samples	I_corr_ (μA/cm^2^)	E_corr_ (V_Ag/AgCl_)	I_pass_ (μA/cm^2^)	E_pit_ (V_Ag/AgCl_)
1 atm, 20 °C	1.54 ± 0.29	−0.252 ± 0.018	13.45 ± 3.54	0.997 ± 0.005
1 atm, 50 °C	2.99 ± 0.37	−0.264 ± 0.022	27.92 ± 3.82	0.936 ± 0.006
1 atm, 90 °C	3.77 ± 0.48	−0.274 ± 0.030	52.37 ± 10.33	0.885 ± 0.008
2 atm, 120 °C	6.90 ± 0.99	−0.301 ± 0.033	120.65 ± 24.21	0.834 ± 0.015
80 atm, 20 °C	1.73 ± 0.41	−0.364 ± 0.021	14.11 ± 3.12	0.999 ± 0.005
80 atm, 50 °C	3.81 ± 0.45	−0.418 ± 0.026	43.26 ± 6.78	0.937 ± 0.005
80 atm, 90 °C	5.71 ± 0.59	−0.467 ± 0.031	86.72 ± 13.94	0.886 ± 0.012
80 atm, 120 °C	11.52 ± 1.63	−0.492 ± 0.034	235.55 ± 41.31	0.812 ± 0.016

**Table 2 materials-16-01313-t002:** EIS fitted results for the AMCs under various temperatures and pressures.

Sample	R_s_ (Ω·cm^2^)	R_c_ (Ω·cm^2^)	CPE_c_ (μF·cm^−2^)	CPE_c_-n	R_t_ (Ω·cm^2^)	CPE_dl_ (μF·cm^−2^)	CPE_dl_-n
1 atm, 20 °C	5.36 ± 0.07	29,506 ± 1887	36.2 ± 1.73	0.78 ± 0.01	362,170 ± 18,027	7.76 ± 0.52	0.69 ± 0.01
1 atm, 50 °C	5.73 ± 0.06	19,397 ± 640	55.3 ± 1.21	0.89 ± 0.01	144,720 ± 9696	9.5 ± 0.57	0.61 ± 0.01
1 atm, 90 °C	3.95 ± 0.05	29.12 ± 1.13	28.9 ± 0.86	0.88 ± 0.01	39,389 ± 472	59.6 ± 0.56	0.83 ± 0.01
2 atm, 120 °C	3.40 ± 0.07	14.29 ± 0.64	14.4 ± 0.96	0.74 ± 0.01	11,357 ± 227	12.3 ± 0.37	0.85 ± 0.01
80 atm, 20 °C	5.54 ± 0.03	21,662 ± 1732	45.8 ± 0.76	0.86 ± 0.01	179,990 ± 13,679	17.3 ± 0.85	0.63 ± 0.01
80 atm, 50 °C	4.94 ± 0.02	16,397 ± 836	56.0 ± 0.59	0.87 ± 0.01	89,779 ± 3129	12.1 ± 0.34	0.65 ± 0.01
80 atm, 90 °C	3.25 ± 0.04	22.94 ± 1.08	32.2 ± 0.89	0.81 ± 0.01	34,556 ± 449	28.4 ± 0.63	0.81 ± 0.01
80 atm, 120 °C	2.30 ± 0.04	4.12 ± 0.21	15.7 ± 0.54	0.84 ± 0.01	10,758 ± 215	15.7 ± 0.66	0.94 ± 0.01

## Data Availability

Data is contained within the article and can be requested from the corresponding author.

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
