# Peer review of "Effects of Temperature and Pressure on Corrosion Behavior of HVOF-Sprayed Fe-Based Amorphous Coating on the Mg-RE Alloy for Dissolvable Plugging Tools"

_materials, 2023, doi:10.3390/ma16031313_

Round 1
Reviewer 1 Report
The paper "Corrosion behaviors of HVOF-sprayed Fe-based amorphous coating on dissolvable Mg-RE alloy under high temperature and high pressure" is suitable for publication in Materials Journal with some minor corrections.
In introduction part : "The normal techniques for fabricating magnesium alloy coatings include chemical conversion [5,6], micro-arc oxidation [7,8], anodizing [9], organic coating [10,11], electroplating and electroless plating [12,13]."; there are more other techniques of coatings for magnesium, like APS: add reference : " In vitro electrochemical properties of biodegradable ZrO2-CaO coated MgCa alloy using atmospheric plasma spraying", Journal of Optoelectronics and Advanced MaterialsVolume 17, Issue 7-8, Pages 1186 - 11921 July 2015.
Please elaborate on the impact of coating elements used by other authors on corrosion resistance.
In materials and methods: please detail more the XRD and SEM parameters.
Please include standard deviation values for tables 1 and 2.
Rest is fine.
Reviewer 2 Report
This is an interesting job that describes behaviour of a coating in a metal alloy in different presion conditions which is not commonly studied. Results provide a new alternative to coatings obtention.
Some details may improve this paper:
SEM images, not specified type of detector used, magnifications, or some microscopy details which can help readers compare images of figure 1 and figure 9 because both SEM images correspond to powders and coatings so preparation and analysis are different. Please complete the information request.
Figure 10 is very interesting and illustrative about the process studied.
Reviewer 3 Report
The manuscript entitled "Corrosion behaviors of HVOF-sprayed Fe-based amorphous coating on dissolvable Mg-RE alloy under high temperature and high pressure" was reviewed. This work and as-obtained results are interesting. In this study, AMCs were prepared on the dissolvable Mg-Gd-Y-Zn-Cu alloy (Mg-RE alloy) via HVOF spraying technology. Under various temperature and pressure, the corrosion resistance of the AMCs and the degradation time of coated Mg-RE alloy were studied systematically. To investigate the mechanism on corrosion behaviors of AMCs, the passive films and corrosion morphologies of AMCs after immersion were also analyzed in details. This manuscript can be accepted for publishing but I have some major remarks before it can be publishable.
1. The abstract is unattractive. It is suggested that the importance of this research work be written in detail.
2. Abstract should have some numerical data.
3. In the Abstract, the authors should emphasize what results the characterizations indicate.
4. The authors should enhance the discussion and comparison with the results in literature.
5. Written is very week. In its current state, the level of English throughout the manuscript needs language polishing. Please check the manuscript and refine the language carefully.
6. Introduction writing part is not satisfactory. The introduction is too long to focus on the major scientific issue. Need to be improved.
7. I have read and evaluated the manuscript and in my opinion the submission does not yet sufficiently justify publication. Discuss the shortcomings of previous work and the gaps and how this work intends to fill those gaps. Related references should be cited:
- Fuel, 332 (2023) 126015.- Environmental Technology & Innovation, 28 (2022) 102947. - International journal of hydrogen energy 42 (8) (2017), 5235-5245; Journal of Industrial and Engineering Chemistry 21 (2015) 1301-1305; Journal of Molecular Catalysis A: Chemical 186 (1-2) (2002) 101-107; Journal of Molecular Catalysis A: Chemical 201 (1-2) (2003) 43-54; Inorganic Chemistry Communications 8 (2) (2005) 174-177
8. The authors should add enough references to support results of their works.
9. Please rewrite the Conclusions. It must be fully supported by the results reported and should include the major conclusions, the limitations of the work and the future work.
Round 2
Reviewer 3 Report
The manuscript entitled "Corrosion behaviors of HVOF-sprayed Fe-based amorphous coating on dissolvable Mg-RE alloy under high temperature and high pressure" was reviewed. In this study, AMCs were prepared on the dissolvable Mg-Gd-Y-Zn-Cu alloy (Mg-RE alloy) via HVOF spraying technology. Under various temperatures and pressures, the corrosion resistance of the AMCs and the degradation time of coated MgRE alloy were studied systematically. After reading authors’ responses, I am grateful for additional information for the backgrounds. However, there are still some minor concepts, I cannot agree with.
Below are some of my specific comments/suggestions:
1. The title is informative and relevant, it could be more specific.
2. The idea of the research seems to be interesting but the set goals are not achieved. What is the innovation of this article? Please highlight it in the article.
3. The Abstract part is weak and it must be more informative by including more mathematical findings and more powerful explanations.
4. The whole generalization for this paper should be given in the introduction.
5. It is suggested that this paper can provide more lists, such as advantages and disadvantages of Mg-RE alloy, which is helpful for readers to quickly obtain information.
6. The authors should add a full and comprehensive discussion to all parts as well.
7. The structure of the introduction needs to be optimized.
8. The structure of the manuscript might need a minor adjustment for a better understanding.
9. Discuss the shortcomings of previous work and the gaps and how this work intends to fill those gaps. Related references should be cited: Journal of Molecular Catalysis A: Chemical 186 (1-2) (2002) 101-107; Journal of Molecular Catalysis A: Chemical 201 (1-2) (2003) 43-54; Chemistry letters 34 (10) (2005) 1444-1445; Journal of Molecular Catalysis A: Chemical 245 (1-2) (2006) 192-199; Journal of Molecular Catalysis A: Chemical 261 (2) (2007) 147-155; Chemical engineering journal 146 (3) (2009) 498-502
